# Adipocyte-Derived Adipokines and Other Obesity-Associated Molecules in Feline Mammary Cancer

**DOI:** 10.3390/biomedicines11082309

**Published:** 2023-08-19

**Authors:** Taylor Marshall, Jing Chen, Alicia M. Viloria-Petit

**Affiliations:** Department of Biomedical Sciences, Ontario Veterinary College, University of Guelph, Guelph, ON N1G 2W1, Canada; tmarshall@nosm.ca (T.M.); jche492@uwo.ca (J.C.)

**Keywords:** feline mammary cancer, obesity, leptin, adiponectin, estrogen, prolactin, serum amyloid A

## Abstract

Obesity has been identified as a serious health concern in domestic cats. Feline mammary cancer (FMC) is also a concern, as it is highly prevalent and aggressive. Considering the identified connection between obesity and breast cancer, it is worthwhile to investigate the potential obesity–cancer relationship in FMC. This review investigated the association between adipokines and other obesity-associated molecules and FMC, with the aim of identifying gaps in the current literature for future research. Based on the reports to date, it was found that tissue concentrations of leptin, serum concentrations of leptin receptor, serum amyloid A, and estrogen correlate positively with FMC, and serum concentrations of leptin correlate negatively with FMC. The roles of adiponectin and prolactin in FMC development were also investigated, but the reports are either lacking or insufficient to suggest an association. Numerous research gaps were identified and could be used as opportunities for future research. These include the need for studies on additional cohorts to confirm the association of leptin/leptin receptor and serum amyloid A with FMC, and to address the role of adiponectin and prolactin in FMC. It is also important to investigate the genetic determinants of FMC, evaluate the use of molecular-targeted therapies in FMC, and exploit the enrichment of the triple-negative immunophenotype in FMC to address current clinical needs for both human triple-negative breast cancer and FMC. Finally, mechanistic studies with any of the molecules reviewed are scarce and are important to generate hypotheses and ultimately advance our knowledge and the outcome of FMC.

## 1. Introduction

Worldwide, obesity has been identified as a serious health concern [1]. In 2016, the World Health Organization estimated that 650 million adults were classified as obese [1]. This epidemic of obesity has serious health consequences, as obese individuals are at increased risk of type II diabetes, cardio-respiratory diseases, osteoarthritis, and various cancers [2].

Breast cancer is a major health concern for women worldwide [3,4]. In the United States, 40,000 women die from breast cancer every year, and about half of patients with metastatic breast cancer that undergo therapeutic treatment experience a relapse within 5 years. The lack of specified treatment and an inability to accurately predict and prevent breast cancer has majorly impacted the care that can be provided to breast cancer patients [4]. Since the human breast is primarily composed of adipose tissue and breast cancer cells come into direct contact with these cells, obesity has been explored as a modulator of breast cancer [4]. Research has determined that obesity is a risk factor for breast cancer in post-menopausal women [4,5]. Obese women are at higher risk for larger breast cancer tumors, lymph node metastasis, and death compared with women who are non-obese [4]. Given the worldwide rate of obesity and its identified role in breast cancer development, it is critical to progress our understanding of the relationship between obesity and breast cancer [4]. 

Obesity is also considered a major health issue in companion animals [2]. In domestic cats of Western countries, obesity has been identified as the most common health disorder [6]. It is estimated that 30–40% of domestic cats are obese, with indoor, neutered cats above the age of 3 being at greatest risk of developing obesity [1,5]. Similarly to humans, obese cats are also at an increased risk of developing health disorders, such as endocrine and lipid metabolism disorders, lower urinary tract disease, gastrointestinal disease, dermatoses, and cancer [7]. 

A cancer of particular interest in cats is feline mammary cancer (FMC). FMC is the third most common neoplasia in cats, making up 12–40% of all tumors identified in cats [5,8]. This type of cancer shares many commonalities with human breast cancer; both diseases show similar molecular classification, epidemiology, and clinicopathological features [3,5,9]. As there is a high degree of similarity between FMC and breast cancer, and since breast cancer is well known to be influenced by obesity, it is important to determine whether obesity also influences the development of FMC [3,4,5,9]. This will aid in improving research design and allow for progress toward common goals that will benefit both species. 

The goals of this review were (1) to examine whether molecular indicators of obesity, such as adipokines, are associated with the development of FMC, and (2) to identify gaps in the current literature in order to develop directions for future research. 

## 2. Obesity Scoring in Domestic Cats

The term obesity is used to describe the excessive deposition of triglycerides in white adipose tissue due to an imbalance in caloric intake and energy expenditure [9]. Obesity in cats can be defined using both gross and molecular scales. Using the gross scale, cats can be classified into underweight, ideal, or overweight categories using the body condition score (BCS) [10]. This tool divides the continuum of superficial body fat into either five or nine categories. In a five-category BCS, cats scoring 4–5 are considered overweight, and in a nine-category BCS, cats scoring 6–9 are considered overweight; obese cats are in the 5/5 or 8–9/9 category [10] (Figure 1). This method is the most common way that veterinarians use to determine whether a cat is overweight or obese, as it only requires a physical examination; however, the molecular scale, which involves examining serum concentrations of proteins and hormones associated with obesity, can also indicate whether a cat is obese [10].

## 3. White Adipose Tissue as an Endocrine Organ 

White adipose tissue is not only the location of triglyceride storage; it is also considered to be an active endocrine organ [2,11,12]. Adipocytes, or fat cells, are known to secrete biologically active molecules, such as proteins and hormones; these are collectively called adipokines. Adipokines are capable of acting in an autocrine, paracrine, or endocrine fashion and have an impact on biological processes, such as inflammation, immune function, metabolism, angiogenesis, cell proliferation, and hematopoiesis [2,11]. Numerous adipokines have been identified; however, we will focus on the ones that have the best-documented role in neoplasia and are primarily secreted by adipocytes, such as leptin and adiponectin [2,11]. Resistin, which was considered an adipokine in the past, is not included as an adipokine in this review, as the evidence indicates that it is mostly secreted by adipose-associated macrophages in humans. In felines, a recent and well-controlled study that included 72 neutered and indoor-confined cats found no association between resistin expression levels and feline obesity when comparing adipose tissue from normal, overweight, and obese cats [13]. An earlier study also observed that circulating levels of resistin are not associated with body weight in cats [14]. 

Leptin is a protein encoded by the *ob* gene and is primarily synthesized and secreted by adipose tissue [11]. The transcriptional regulation of leptin is dependent on the energy flux within adipocytes, which means that leptin production is increased during a postprandial or fed state and decreased during fasting [11]. Since leptin is primarily produced in adipose tissue, the amount of fat tissue in an individual is proportional to the amount of leptin within the body [5,11]. Leptin acts by binding to leptin receptors (Ob-R) in the satiety center of the brain, which is located in the hypothalamus, to cause appetite suppression [11]. Thus, after the ingestion of food, leptin is secreted by adipose tissue to cause a feeling of fullness [11]. Failure to produce leptin or the development of leptin resistance can contribute to the development of obesity due to the body’s inability to maintain energy homeostasis [11]. Obesity can also occur if there is a low concentration of leptin receptors or if the leptin receptor is nonfunctional, as this molecule regulates the ability of leptin to cause physiological changes in the body [11]. 

The adipokine adiponectin also contributes to the development of obesity [11]. Adiponectin is a protein whose gene is one of the most highly expressed by adipocytes [11]. In contrast to leptin, the amount of fat tissue in the body is inversely correlated with the amount of adiponectin [11]. Adiponectin secretion is stimulated by insulin, and adiponectin’s primary action is to increase insulin sensitivity and glucose uptake into cells by promoting the translocation of glucose transporters to the cell surface and increasing glycolysis and fatty acid oxidation [11]. In obese individuals, adiponectin levels are decreased, which may occur due to the increased production of pro-inflammatory cytokines that act as inhibitors of adiponectin gene expression [11]. Furthermore, high insulin levels downregulate adiponectin receptors; therefore, hyperinsulinemia, which is associated with obesity, can lead to adiponectin resistance by modifying adiponectin receptor availability [11]. 

Due to their diverse roles and their capability to act throughout the body, adipokines have been implicated in the pathophysiology of many diseases [15]. When the body is in an obese state, the dysfunctional secretion of adipokines from increased adipose tissue leads to an imbalance in metabolism and can influence the development of disorders such as mammary cancer [5,15]. Increased leptin secretion in an obese state contributes to mammary gland development and cellular proliferation [5,15]. Additionally, although leptin is mainly secreted by adipocytes, it can also be expressed by pathologically altered cells, such as cancer cells [5]. In the case of adiponectin, decreased adiponectin secretion due to obesity can cause hyperinsulinemia, which is associated with neoplastic growth, cell proliferation, and metastasis [5,15]. Due to these findings, leptin and adiponectin have been identified as potential molecular mediators of FMC [5].

## 4. Obesity as a Chronic Inflammatory Disorder

Obesity can be described as a nutritional disorder, where there is an imbalance in energy intake and expenditure, but it is also known to be a chronic systemic inflammatory disorder [2,12]. Increased adipose tissue leads to dysfunctional production and the secretion of adipokines, such as an increase in pro-inflammatory leptin and a decrease in anti-inflammatory adiponectin [10]. The result is chronic, low-grade inflammation that triggers the acute-phase response of the innate defense system and the secretion of acute-phase proteins [10,16]. 

Acute-phase proteins (APPs) are secreted in response to pro-inflammatory cytokines/adipokines and have immunomodulatory activity in order to protect tissues from inflammation, infection, or trauma [16]. An APP of particular interest is serum amyloid A (SAA), which works to protect tissues from an inflammatory stimulus by inhibiting free radicals [16]. SAA is recognized as a sensitive biomarker for inflammation in cats, as feline SAA concentrations increase dramatically in response to inflammation [1,16]. This protein can also be used to identify feline obesity since SAA concentrations were shown to be elevated in cats with accumulated visceral fat [1]. Although acute-phase proteins are conventionally thought to be produced and secreted by the liver, adipocytes and inflamed mammary glands are also capable of producing SAA [17]. These findings make SAA a molecule of interest, as it could be a biomarker for FMC and possibly mediate FM carcinogenesis and/or cancer progression [17,18].

## 5. Sex Hormone Modulation of White Adipose Tissue 

The role of adipose tissue as an endocrine organ also impacts the body’s reservoir of sex hormones by modulating circulating levels and relative ratios of estrogen and prolactin [2,11,12,19]. Estrogen contributes to obesity by influencing the synthesis of leptin, thus affecting the regulation of appetite and fat deposition [6,12,20]. In female cats, estrogen production primarily occurs in the ovaries [12]. Once a cat has undergone an ovariohysterectomy (removal of ovaries and uterus), which occurs when a female cat is spayed, it was shown that female cats increase in body weight by 16% and increase by one point on the BCS due to increased food intake [6,12,21]. This occurs because adipocytes have estrogen receptors that modulate leptin synthesis [20]. When adipocyte estrogen receptors are activated, leptin secretion increases, and thus, when a cat’s ovaries are removed and estrogen secretion is decreased, less leptin is produced [20]. This increases the risk of feline obesity due to a lack of appetite suppression by leptin [20,21]. 

Although estrogen is primarily produced in the ovaries, adipose tissue also serves as a secondary site of estrogen production [4]. Adipose tissue possesses the enzyme aromatase, which converts androgens or estrogen precursors into estrogen [4,22]. As adipose tissue is in direct contact with mammary tissue, it was found that adipokines are capable of stimulating aromatase production, and thus, elevating the local estrogen levels in mammary tissue [4]. Studies demonstrated the role of estrogen in normal and neoplastic feline mammary tissue growth as a contributor to the proliferation of mammary epithelial tissue [23]. Estrogen is a mitogen for epithelial cells of the mammary gland by stimulating G_0_/G_1_ resting cells to re-enter the cell cycle and complete a round of cell division [4]. This means that extensive exposure to estrogen due to adipocyte-mediated aromatization can cause hyperproliferation of mammary epithelial tissue, which can then progress to a preneoplastic state and eventually lead to the development of a mammary carcinoma [4,23]. Thus, estrogen was identified as a possible contributor to FMC development [23]. 

Prolactin (PRL) is a hormone secreted from the anterior pituitary gland and whose role pertains to reproduction, lactation, mammary gland development, immune response, and angiogenesis [19,24]. It has also been associated with the development of obesity [19]. In cats, as well as in humans, high serum concentrations of PRL were shown to correlate positively with obesity [19]. This occurs, at least in part, because prolactin acts on the hypothalamus to stimulate appetite, and thus, increased PRL levels promote the maintenance and production of adipose tissue [25]. Since PRL is primarily associated with the mammary gland and its function, it has also been recognized as a potential biomarker for FMC [24]. In canines, rodents, and humans, PRL was shown to be produced by mammary tissue and was also identified as an important molecule in the initiation of mammary neoplasia [24]. PRL is known to stimulate mammary epithelial cell proliferation and differentiation in cats, which invites further investigation into the association of PRL with FMC [24].

## 6. Feline Mammary Cancer

Mammary cancer is highly prevalent in domestic cats [5]. It is the third most common cancer in cats, comprising 12–40% of diagnosed tumors in this species [5,8]. Risk factors for FMC include age, with the incidence of FMC increasing significantly after the age of 6 and peaking at 10–11 years of age [26,27]. In addition, cats of the Siamese breed were reported to be at a twofold higher risk of developing FMC when compared with all other breeds [28]. The diagnosis age for female Siamese cats also tends to be younger compared with other breeds [28]. These findings suggest a genetic link to FMC development and the possibility of Siamese cats serving as a model for the genetic risk of breast cancer [27,29]. It is important to add that according to studies based on the records of cats seen at veterinary practices in England, Siamese cats do not seem naturally predisposed to obesity [30] or have shorter longevity than mixed-breed cats [31]. However, it remains to be determined whether Siamese cats are more prone to specific subtypes or more aggressive forms of mammary cancer. 

In general, FMC is aggressive, with approximately 85–95% of tumors found to be malignant (adenocarcinomas); grow rapidly; and metastasize to regional lymph nodes, the lungs, the liver, and the pleura [8,32,33]. Cats with disseminated tumors are classified as stage IV and have a poor prognosis, with a median survival time of 1–3 months, with asymptomatic cats showing a better prognosis [33]. Metastatic disease is the primary cause of death in FMC, and the role of adjuvant therapy at this stage has not been established. A recent retrospective study by Petrucci et al. suggests that primary tumor surgery can significantly extend the survival times of cats with metastatic mammary cancer, and although not significant, metronomic chemotherapy and toceranib phosphate have the potential for improving the overall survival of cats with metastatic cancer and were associated with better quality of life [33].

FMC can be divided into several subtypes based on the molecular expression of the tumour [8]. Broadly, FMC can be divided into luminal types, the HER2-positive type, and the triple-negative types. These subtypes are very similar to the subtypes of breast cancer, which make FMC of interest as a potential model for breast cancer [3]. However, it must be noted that the mechanisms responsible for specific molecular subtypes might be different in cats compared with humans. For instance, in HER2-positive human breast cancer, gene amplification is generally responsible for HER2 overexpression. In contrast, in FMC, mRNA overexpression in the absence of gene amplification might be responsible for HER2 protein overexpression [34,35]. Also important and noted by Soares et al. [35], the proportion of HER2-positive FMC reported varies greatly between studies by different groups. Based on this, this group optimized immunohistochemistry (IHC) and in situ hybridization (ISH) methods for fHER2 determination [35]. In this study, antibody A0485 (1:300) with citrate buffer antigen retrieval in a water bath at 95 °C for 60 min yielded the best results, which were reproduced in subsequent studies [8]. It would be wise to use this protocol in routine molecular phenotyping of FMC and in future research to ensure reliability and reproducibility.

The luminal subtypes of FMC can be further divided into type A and type B [8]. Patients with type A tumors are known to have the best prognosis of all of the FMC subtypes, which is also recognized in human breast cancer patients [8]. This subtype is characterized by tumors with overexpression of estrogen and/or progesterone receptors, low Ki-67 index, and the absence of human epidermal growth factor receptor 2 (HER2) [8,36]. Type B tumors are the most common subtype of FMC and can be separated further into two groups: HER2 negative and HER2 positive [8]. Both subtypes of luminal B FMC have estrogen and progesterone receptor expression, but they differ in that luminal B HER2 negative tumors have high Ki-67 indices and luminal B HER2 positive tumors have an overexpression of HER2 receptor, as well as having higher malignancy [8,36]. Luminal B subtypes are known to be more aggressive than luminal A but not as aggressive as the non-luminal subtypes, which is also true for human breast cancer of the luminal B subtype [8]. 

The non-luminal subtypes of FMC are made up of HER2-positive and triple-negative [8,37]. The HER2-positive subtype of FMC is designated by the overexpression of HER2 and lack of estrogen and progesterone receptors [8,36]. Compared with the luminal types, HER2-positive FMC shows high proliferation and is identified as being associated with the second-lowest disease-free interval and survival time of the FMC subtypes [36,38,39,40].

The triple-negative subtype of FMC can be identified due to the absence of estrogen, progesterone, and HER2 receptors [8,36]. Tumors of this subtype can be divided into basal-like, which is positive for cytokeratin 5/6 (CK 5/6) expression, and normal-like, which has no associated molecules [8,36]. Triple-negative subtypes of FMC are known to be highly aggressive and correlated with a poor prognosis, which are also features of human triple-negative breast cancer [8,9]. These common features in humans and cats make this subtype of FMC of particular interest as it may serve as an effective model for triple-negative breast cancer and may allow for the development of targeted therapies [8,9]. Table 1 summarizes the data described above. To avoid confusion with regard to the Ki-67 index presented in this table for specific molecular subtypes, it must be clarified that although a low Ki-67 is a common feature of luminal A FMC, other molecular subtypes may also include tumors with a low Ki-67 index. For instance, a low Ki-67 index was found in 52.6%, 27.5%, and 22.7% of luminal B HER2-positive, triple-negative basal-like, and normal-like subtypes, respectively, in a thorough study by Soares et al. [8]. 

Finally, it is important to note that in FMC, ER, PR, HER2, and Ki-67 are not evaluated in routine histopathology. However, research on immunophenotypes has shown that the most common FMC subtypes are luminal B HER2-negative and the triple-negatives. Only the luminal A type is associated with a better outcome, and although there is limited data on the predictive role of FMC immunophenotypes, recent evidence suggests that the HER-2-positive type may respond well to HER-2-targeted therapy [41]. 

## 7. Mediators of Obesity and their Association with FMC

As previously identified, there are several molecular mediators of obesity that also contribute to cancer development. Some of the proteins and hormones that are of interest due to the existing evidence of their role or potential role in FMC include leptin and leptin receptor, adiponectin, SAA, estrogen, and prolactin.

### 7.1. Leptin

Leptin and leptin receptor are well-documented mediators of obesity through their role in appetite suppression [11]; however, their potential role in FMC was only recently explored [5]. Gameiro et al. assessed leptin and leptin receptor in FMC cells and associated stroma, as well as in serum [5]. Within mammary cancer cells and neighboring adipocytes, leptin was shown to be overexpressed. Inversely, serum leptin was shown to be decreased in cats with FMC, which is a trend that has also been noted in pre-menopausal breast cancer patients [5]. Of the FMC subtypes, luminal B and triple-negative subtypes were observed to have the greatest tumor expression of leptin, while luminal B and HER2-positive subtypes were found to have the lowest serum leptin levels [5]. Since triple-negative subtypes of FMC were shown to have both high tumor and serum expression of leptin, this suggests that leptin is associated with more aggressive and proliferative mammary carcinomas, which was also reported in human breast cancer [5]. Levels of serum leptin receptor were shown to be elevated for all FMC tumor subtypes [5]. High levels of serum Ob-R are also associated with smaller tumor sizes, which suggests that leptin receptor shedding may occur in small tumors and larger tumors maintain higher leptin receptor expression [5]. Therefore, the authors hypothesized that higher levels of leptin receptor on a tumor, and thus, greater ability for leptin to act upon cancer cells, are associated with FMC tumor growth and survival [5]. Further studies with FMC cells in vitro and in vivo are necessary to address this hypothesis.

### 7.2. Adiponectin

The role of adiponectin in FMC remains to be determined. In humans, adiponectin is known to be anti-inflammatory and anti-atherogenic, as it suppresses TNF-α production by macrophages, as well as the movement of monocytes into subendothelial areas, which inhibits the development of atherosclerotic plaques [11]. These effects may also provide protection against cancer cell growth by suppressing the angiogenesis associated with tumour formation [11]. Low adiponectin levels are observed in humans with a variety of cancers, including breast cancer, as well as in obese humans, which supports the statement that obesity is a risk factor for the development of breast cancer [4,5,11]. It was noted that feline and human adiponectin share strong homology and are both decreased in obese individuals [11]. In a study by Okada et al., lower levels of adiponectin in cats were found to be associated with a BCS of >7/9 and accumulated visceral fat, and were considered an indicator of obesity disease [1]. These findings are supported by a recent study that examined adiponectin expression using quantitative real-time PCR, which found a negative association between obesity and adiponectin expression in feline subcutaneous adipose tissue [13]. Our review of the literature evidenced no studies investigating levels of adiponectin in association with FMCs. Therefore, future studies aiming to identify molecular links between obesity and FMC should evaluate adiponectin in tumors and surrounding adipocytes, as well as serum. This would identify associations, if any, with cancer development, the cancer type, the disease-free interval, and overall survival, that may suggest a role of this adipokine in FMC. Based on these findings, the biology should be investigated.

### 7.3. Serum Amyloid A

SAA, which is an acute-phase protein and potent indicator of acute or chronic inflammation conditions, has been identified as a biomarker for inflammation and obesity in felines, and may also be a mediator of FMC [1,18]. Assessment of APP levels at diagnosis in the serum of cats with mammary carcinoma (*n* = 50) versus clinically healthy controls (*n* = 12) suggested that the development of spontaneous FMC is associated with an APP response and oxidative stress given that significant changes in APPs, including SAA and haptoglobin (Hp), were found in diseased cats compared with controls [42]. A significant positive association was found between serum levels of SAA and Hp and the following: tumour ulceration, neoplastic emboli in lymphatic vessels, regional lymph nodes and distant organ metastasis, histological type and grade, necrosis, and higher proliferative activity [42]. A potential role of SAA in metastasis is supported by an in vitro study by Tamamoto et al. [18], which suggests that SAA contributes to tumor cell invasion in FMC by stimulating matrix metalloproteinase-9 (MMP-9) production [18]. MMP-9 is part of a family of proteases that are able to degrade the extracellular matrix and basement membrane, thus facilitating dissemination. In this study, three out of four FMC cell lines showed increased expression of MMP-9 following exposure to SAA, and this correlated with increased invasiveness in a transwell assay [18]. Taken together, the above studies suggest that SAA and Hp should be further investigated as biomarkers for diagnosis, prognosis, and monitoring of local/distant recurrence in FMC. Further, close monitoring of obese female cats with high levels of APPs might be helpful for the early detection of metastatic FMC, hopefully contributing to improved outcomes. Studies on mice are necessary to demonstrate the role of SAA in tumourigenesis and/or metastasis in vivo.

### 7.4. Estrogen

Estrogen is known to be a mediator of obesity in cats and a contributing factor to the development of FMC [23]. Prolonged exposure to estrogen was shown to induce the proliferation and accumulation of genetic errors in mammary epithelial cells, which can lead to the development of neoplasia [23]. Studies also showed that an ovariohysterectomy can be used as a protective procedure against FMC, as it reduces the risk of FMC development by 91% if the procedure occurs before 6 months of age and 86% if the procedure occurs before 1 year of age [23]. Additionally, the use of aromatase inhibitors in postmenopausal breast cancer patients was shown to be effective, as these medications block the conversion of androgens into estrogens by aromatase in adipose tissue, which is the main source of estrogen in a postmenopausal patient [4,22]. These factors indicate that the presence of estrogen is positively correlated with mammary neoplasia in both felines and humans. However, immunohistochemical analysis of FMC tumors has shown that a minority of them, ranging from 8% to 20%, are positive for ER [43], while the most aggressive subtypes of FMC are HER2-positive and triple-negative, which lack estrogen receptors [8,9]. Apart from suggesting that aggressive sub-types of FMC would be unresponsive to hormonal therapies that target estrogen receptors or estrogen production, these facts invite further research into the role of obesity-associated estrogen production in the development of FMC, specifically in spayed cats.

### 7.5. Prolactin

PRL levels were also shown to be associated with feline obesity and may contribute to FMC [24]. As mentioned above, PRL may promote the expansion of adipose tissue and obesity indirectly by increasing appetite [24]. However, there is evidence to suggest that PRL also promotes the accumulation of adipose tissue directly by facilitating the differentiation of adipocytes [44]. Although PRL is mainly produced by the pituitary lactotroph cells, it can also be synthesized by extra-pituitary sources, such as the mammary gland, the uterus, the lymphocytes, and the adipose tissue [44]. Further, the release of PRL by adipose tissue is negatively regulated by insulin and varies depending on the location, differentiation state, and type of adipose tissue, as well as on body mass index (BMI) [44]. A study by Hugo et al. [45] showed that in cultured explants from obese human patients, subcutaneous adipose tissue releases more PRL than visceral adipose tissue, but this difference by location of fat depot was not seen with cultured explants from lean patients. Further, adipose explants from lean patients release more PRL than those from obese patients. A negative relationship was observed between the PRL levels released by explants and the patient’s BMI, but no association was found between serum PRL levels and BMI [45]. Serum PRL levels in male cats were found to increase 8 weeks following castration and this paralleled the increase in body weight [25]. This is in agreement with the role of this hormone in adipose tissue deposition. Our revision of the literature to date found no studies to suggest an association between circulating levels or adipose tissue levels of prolactin and FMC. However, prolactin expression was previously assessed in FMC tissue. Trummel et al. [24] analyzed PRL expression by IHC in mammary tissue samples from eight cats, including two benign hyperplasia, two adenoma, and four adenocarcinoma cases. PRL immunolabelling was only positive in two out of the four adenocarcinomas, specifically in neoplastic mammary epithelial cells located in heterogeneous clusters [24]. The cats with mammary PRL-positive tumors were noted to have greater survival time than those that were negative; however, the sample size was too small to establish any conclusions. Further research is required to determine whether tumor cells’ prolactin levels can be used as prognostic marker for FMC [24]. Table 2 summarizes the data discussed above.

## 8. Conclusions and Future Directions

In this review, potential molecular mediators of FMC associated with obesity are discussed, including leptin, adiponectin, serum amyloid A, estrogen, and prolactin. FMC and its subtypes are also discussed to further our understanding of how these molecular mediators may favor specific subtypes of FMC. The existing literature indicates that our knowledge of the association between obesity and FMC is very limited. 

The reports to date indicate that leptin is overexpressed in FMC, with the highest in luminal B and triple-negative tumors, and lower in circulation, which supports the idea that leptin is recruited into cancerous mammary tissue to facilitate tumor growth [5]. Regarding adiponectin, there was no research on its role in FMC by the time this review was completed. Serum amyloid A was found to be associated with several parameters of poor prognosis, including local and distant tumor dissemination [42], and in vitro studies suggest a role in promoting tumor invasion [18]. Excessive exposure to estrogen in felines was linked to hyperproliferation and neoplasia of the mammary gland, with spaying of female cats before 6 months of age resulting in the best available protocol for FMC prevention [19,23]. Finally, prolactin expression has only been assessed in feline mammary tissue in a study with a very small sample size, where it was shown to be positive only in FMC compared with benign lesions and cats with positive tumors (*n* = 2) survived longer [24]. All findings discussed are summarized in Figure 2. The reports for all the molecules reviewed suggest their potential as predictive and prognostic biomarkers in FMC, but they are very limited and need confirmation in independent cohorts. Mechanistic studies are scarce and necessary to generate ideas for translational research. 

Another important area that will benefit from research is FMC genetics. In humans, a clear genetic link was established for breast cancer [46]. Mutations in the *BRCA1* and *BRCA2* genes were shown to be associated with a risk of 60–85% of developing breast cancer [46]. A genetic link to the development of FMC was also proposed, as cats of the Siamese breed are known to be at increased risk of FMC development, but the gene or genes responsible for this increased risk have not been identified [27,29]. There are very few studies on the predisposition of cats to FMC, and the ones reported to date have only examined specific genes, such as *BRCA1/2* and *TWIST1*. For instance, two *TWIST1* intronic germline variants were identified in a study that included 34 cats with FMC (3 cases with variant 535delG and 4 with variant 460C >T), but they did not affect mRNA expression [47]. Of two studies on *BRCA1/2*, one showed no variants in 24 cats with mammary carcinoma [43], while the other identified four germline variants of *BRCA1* (intron 9) in three out of the nine FMC-bearing cats genotyped [48]. Sequencing malignant and normal cells from Siamese cats and other cat breeds would aid in identifying gene variants predisposing cats to FMC development. Identifying the genetic influence on FMC would help to improve the incidence of FMC, as cat owners could participate in genetic testing for these genes and could take protective measures if their cat carried a predisposing mutation, such as neutering their cat prior to 6 months of age, and close monitoring for the early detection of mammary tumors.

Finally, the current therapy of FMC is sub-optimal but there are opportunities for improvement. Although FMC and breast cancer have similar subtypes that express the same distinctive molecules, FMCs are not immunophenotyped routinely and molecular-targeted therapies are not currently used [3,41]. Treatment for FMC could be improved by completing IHC on FMC tumors to determine the molecular subtype and decide on appropriate therapy, if available. For this, it is imperative to demonstrate the efficacy of molecular-targeted therapy approved for ER-positive and HER2-positive breast cancer. Identifying cost-effective molecular-targeted therapies to treat triple-negative FMC, which is one of the most common and aggressive types of FMC, is imperative. Establishing clear associations between obesity-associated molecules and triple-negative FMC (if any) could be a way to identify new molecular targets for this aggressive type of cancer for both cats and humans. Further, the similarity in the physiology and endocrinology of obesity in felines and humans makes domestic cats suitable subjects for further investigation of the effects of obesity on breast cancer development.

## Figures and Tables

**Figure 1 biomedicines-11-02309-f001:**
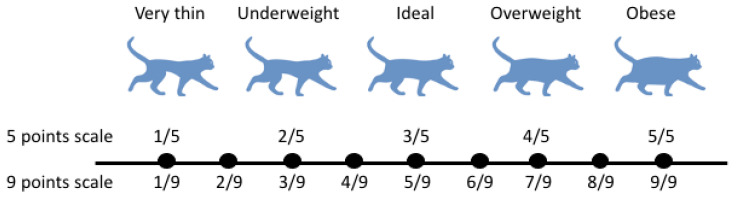
Body condition score for cats (5- and 9-point scales included). Cat Image obtained with licence from Adobe stock (Copyright © 2023, Creative Cloud, Adobe Inc., San Jose, CA, USA).

**Figure 2 biomedicines-11-02309-f002:**
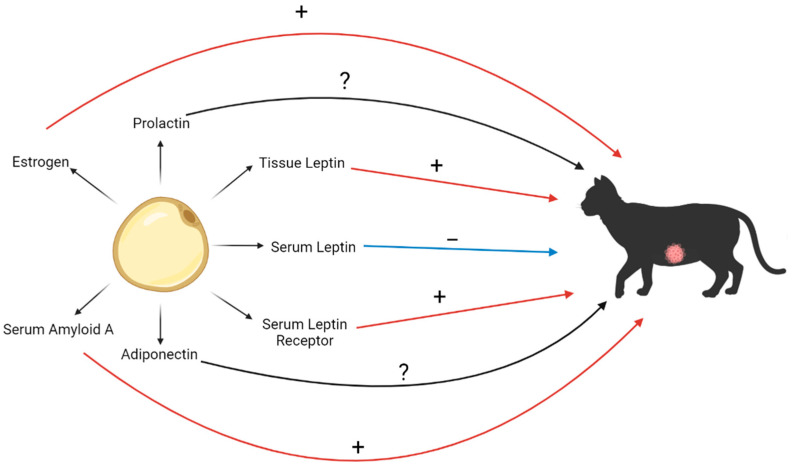
Schematic of associations between molecular mediators of obesity and FMC.

**Table 1 biomedicines-11-02309-t001:** Molecular subtypes of feline mammary cancer (FMC). Summarizes data from reference [8], unless otherwise indicated.

Subtypes of FMC	Associated Molecules	Clinical-Pathological Features	Prognostic and/or Predictive
Luminal A	ER and/or PR Low Ki-67 index	Less invasive and proliferative tumors Low biologic aggressiveness	Longest survival time and disease-free interval
Luminal B HER2-negative	ER and/or PR High Ki-67 index	Less aggressive than luminal B HER2-positive	Better prognosis than the HER2-positive type
Luminal B HER2-positive	ER and/or PR fHER2	High biologic aggressiveness	May benefit from HER2 targeted therapy
HER2-positive	fHER2	Highly proliferative	Second-shortest survival time and disease-free interval [36,38] May benefit from HER-2 targeted therapy [41]
Triple-negative basal-like	CK 5/6	Associated with tissue necrosis High biologic aggressiveness Large tumors	Shortest survival time and disease-free interval No molecular-targeted therapy identified
Triple-negative normal-like	None	Associated with tissue necrosis High biologic aggressiveness Large tumors	

ER: estrogen receptor; PR: progesterone receptor; fHER2: feline human epidermal growth factor receptor 2; CK5/6: cytokeratin 5/6; Ki-67: Kiel clone 67, which is a marker of proliferation.

**Table 2 biomedicines-11-02309-t002:** Obesity-associated molecules and their correlation with FMC.

Molecule	Correlation with FMC	Reference
Tissue leptin	Positively correlated	[5]
Serum leptin	Negatively correlated	[5]
Serum leptin receptor	Positively correlated	[5]
Adiponectin	None documented to date	N/A
Serum amyloid A (SAA)	Positively correlated	[42]
Estrogen	Positively correlated	[23]
Prolactin	Unclear	[24]

## Data Availability

Not applicable.

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
