# Peer review of "Adipocyte-Derived Adipokines and Other Obesity-Associated Molecules in Feline Mammary Cancer"

_biomedicines, 2023, doi:10.3390/biomedicines11082309_

Round 1

Reviewer 1 Report

The manuscript submitted by Marshall et al. entitled "Adipocyte-derived adipokines and other obesity-associated molecules in feline mammary cancer" deeply reviewed the putative oncogenic effects of the adipocyte-tissue in feline mammary carcinoma. The article is well-written, structured and organized.  The article covers many aspects of adipose and the majority of the relevant references are cited. However the following references should be added at:

Line 238 - 10.3390/vetsci8080164

Line 241 - 10.1038/s41598-020-60860-3; 10.3390/cells11162578

Author Response

We appreciate your positive review. We added the suggested references; they are references #37 and #39 in the revised manuscript.

Reviewer 2 Report

Comments on the manuscript biomedicines-2515976-peer-review-v1 entitled “Adipocyte-derived adipokines and other obesity-associated molecules in feline mammary cancer” by Marshall et al.

The manuscript is an interesting review on the potential effect of adipokines on feline mammary tumorigenesis. New articles on this theme are welcome. However, there are some minor changes that should be amended.

Line 86: In the sentence “…called adipokines . Adipokines …” there´s an extra space that should be removed.

Lines 209-210: Please replace “…85-95% of tumors being malignant adenocarcinomas that grow…” to “…85-95% of tumors being malignant (adenocarcinomas) that grow…”, as adenocarcinoma is a malignant epithelial tumour. 

Lines 236: In the sentence “Luminal B subtype[8].” there´s a space missing.

Line 244: Please amend the following sentence “…cytokeratin 5/6 expression…” to “…cytokeratin 5/6 (CK 5/6) expression…”

Lines 245-246: In the sentence Triple-negative subtypes of FMC are known to be highly aggressive, malignant, and are correlated…” remove malignant from the sentence, as a carcinoma is a malignant tumour.

Table 1: Please replace “Low malignancy” and “High malignancy” for “High” or “low biologic aggressiveness”. Insert a legend as a table footer, of ER, PR, HER2, CK5/6, Ki-67.

Line 285: Put “in vitro” and “in vivo” in italic.

Line 321: Put “in vitro” in italic. 

There´s an extra space in the sentence before the comma…” Tamamoto et al [18] ,”. Please amend.

REFERENCES

There are some journal names in full and others in short (as ref 10, and 23), and missing pages (ref 10, 13, 24, 36, 42). Please amend.

Ref 24- please amend “Adenomas and Ade‐ nocarcinomas…”

Author Response

Many thanks for your positive review and for the suggested edits. They have all been incorporated and can be tracked throughout the text. We went over the references and incorporated missing information. For those journals that recommend citing article number instead of page number, we enter article number manually, as the citation format was not compatible with including article number by our reference manager.

Reviewer 3 Report

The authors from the manuscript entitled “Adipocyte-derived adipokines and other obesity-associated molecules in feline mammary cancer” written a review paper about the obesity associated molecules with feline mammary cancer. However, their performed a simple literature review. The review should be made as a meta-analysis and a critical review providing more specific information regarding the obesity and feline mammary gland tumors. In the way presented, do not increase the knowledge of thew specific area. Moreover, this reviewer found a serious misconduct in the manuscript citation, with authors citing their own university instead citing the original manuscripts. Is not a problem the citation of their own institution group since the original ones are cited. The authors used data from an original manuscript and provided a citation of a literature review instead the original paper. There is no reason for this option, since the original paper could be easily assessed by authors.  Please, see other comments below:

1.     It is a simple literature review with some parts authors claimed that no studies were performed in cats (line 289). Since it is unclear in cats, no conclusion could be made on this association.

2.     Authors performed a literature review of a specific topic and were no deep on their search. They opted to cite literature reviews instead original manuscript with the research data. i.e. Govoni et al. 2021.

3.     Figure 1 quality should be increased

4.     Regarding BRCA1 and BRCA2 mutation in cats with mammary gland tumors, the authors performed a tremendous flaw. Between lines 429 up to 446, authors described results for a manuscript without citing the original manuscript. The manuscript describing the 3 out of 9 nine FMC and the study investigating 24 cases of FMC were not cited properly. The authors opted to cite a manuscript that reviewed both manuscripts. The paper cites (that reviewed both previous manuscripts) were from the same institution (University of Guelph). Thus, authors opted to cite their colleagues instead the original manuscripts.

5.     I strongly advice authors to not do this type of citation because can seems as an ethical misconduct.

Author Response

Thanks for your review of our manuscript. Because of the limited data on the subject, we considered a standard literature review more suitable than a meta-analysis. The review includes data that was not in previous reviews, assesses this data in relation to prior literature, and provides specific ideas for future research to advance the field. In this sense, we believe it advances the field. However, based on your comments, we have expanded on some topics, which are specified below in our responses to your specific points:

  1. We apologize for the misunderstanding regarding our approach to the section of adiponectin. We did not find studies that specifically link adiponectin levels to FMC, and we did not make any conclusions on this specific association. Thus, in the section on adiponectin, we only mentioned reported associations between low adiponectin levels and feline obesity, as well as similarities found between humans and cats in the context of the adiponectin-obesity connection. We considered these points important to support our statement on the need to do research on adiponectin in the context of FMC.

  2. We appreciate your comment. The reasons for citing reviews, was to give credit to the authors that have reviewed the topic before us. However, based on your comment, we have now cited and discussed the original manuscripts when appropriate (see lines 457-462), such as was the case for Govoni et al (doi: 10.1111/vco.12685), which is now reference #48.

  3. Figure 1 was updated and the quality has improved.
  4. We included the original references, which are refs.# 43, 47 and 48 in the revised manuscript. We also made edits to the second and third paragraph of the "Conclusions and Future Directions" section of the manuscript, which we believe describe the data better.

  5. Thanks for the advice. However, it is important to mention that there is no rule that precludes the citation of authors from the same institution, if their work is pertinent to the subject matter being discussed. Citing the review was not "serious misconduct" but rather a way to acknowledge authors that have reviewed the subject before us, as we did with with other authors in others sections of the manuscript.

Reviewer 4 Report

Dr Taylor Marshall and co-workers have written an exhaustive review on current knowledge of the correlation of obesity/molecular signalling in adipose tissue and the development of mammary tumors in cats. They also add important and comparative information to the similar situation in human breast cancer. The review is in general well written and adds value into this rather new research field in FMC. I have some minor comments that would like the authors to address.

General:

The HER-2 analysis is fundamental to achieve a complete phenotyping of the FMC and enable further comparative research with human BC. The authors are not commenting the need of selecting correct and validated methods for cats, as the reports of HER-2 expression have varied significantly depending on analysis made. Therefore, it may be difficult to use some older references as proofs for either low or high expression of HER-2 in FMC. I would like the authors to add a comment on this in the section of HER-2 or in the discussion. If the authors have found a better validated feline HER-2 method during the literature review it could be good if you could promote that.

The current focus on predisposition of cancer, including human BC, in obese humans are very much revolving around the chronic inflammatory situation that different signals in fat tissue creates and the cross-talk with the immune system. In the current review, this is absolutely addressed, although the endocrine relations are more heavily described. I would welcome if the authors (at least) can add a summarizing table, or a figure, that also adds the complexity of how fat tissue promotes a systemic inflammation/chronic inflammatory state that promotes carcinogenesis.

Specific:

Ln 217: Change ”no” to not.

Table 1: It looks like it would be practically possible to identify Luminal A, only based on investigate Ki67. IF Ki67 is low, this automatically puts it as Luminal A. If that is the case Ki67 should be recommended to be added as a routine investigation in FMC, as identifying Luminal A Y/N would add significant value in the clinical situation as Luminal A is the only subtype with rather good prognosis. Please elaborate on this in the text or add explanation that this is not the case and that Ki67 can be high/low in other subtypes.

Table 2: Excellent.

Some minor spell check would be good. Otherwise, English is acceptable.

Author Response

Thanks for assessing our manuscript and your suggestions.

In response to your first general comment, we added a short discussion on the assessment of fHER2 expression in FMC, which can be seen in lines 224 to 236 of the revised manuscript.

With regard to your second general comment, which suggests the addition of a table or figure that summarizes the complexity of the connection between inflammation and carcinogenesis, we believe this is beyond the scope of the manuscript. It will be difficult to add a table or figure on this subject without providing an extensive explanation in the manuscript. In addition, such section would need to focus on data in humans, as the connection between inflammation and carcinogenesis, specifically with regard to feline mammary cancer, is poorly documented. We hope you understand these reasons, and we look forward to explore the subject in future research.

In response to your specific comments, we have made the minor edit suggested, checked the manuscript for minor spelling mistakes and corrected them. We have also added a few lines, to clarify the issue of low ki-67 index in other FMC molecular subtypes, apart from Luminal A (lines 264-269).

We are happy to know that you found Table 2 informative. We appreciate your positive comment on this component of the manuscript.